# ADDITIVE MARGIN SOFTMAX
# FOR FACE VERIFICATION

**Feng Wang**
Department of Information and Communication Engineering
University of Electronic Science and Technology of China
Chengdu, Sichuan 611731 China
`feng.wff@gmail.com`

**Weiyang Liu & Hanjun Dai**
College of Computing
Georgia Institute of Technology
Atlanta, United States.
`{wyliu, hanjundai}@gatech.edu`

**Haijun Liu & Jian Cheng**
Department of Information and Communication Engineering
University of Electronic Science and Technology of China
Chengdu, Sichuan 611731 China
`haijun_liu@126.com chengjian@uestc.edu.cn`

## ABSTRACT

In this paper, we propose a conceptually simple and geometrically interpretable objective function, i.e. additive margin Softmax (AM-Softmax), for deep face verification. In general, the face verification task can be viewed as a metric learning problem, so learning large-margin face features whose intra-class variation is small and inter-class difference is large is of great importance in order to achieve good performance. Recently, Large-margin Softmax (Liu et al., 2016) and Angular Softmax (Liu et al., 2017a) have been proposed to incorporate the angular margin in a multiplicative manner. In this work, we introduce a novel additive angular margin for the Softmax loss, which is intuitively appealing and more interpretable than the existing works. We also emphasize and discuss the importance of feature normalization in the paper. Most importantly, our experiments on LFW and MegaFace show that our additive margin softmax loss consistently performs better than the current state-of-the-art methods using the same network architecture and training dataset. Our code has also been made available[1].

## 1 INTRODUCTION

Face verification is widely used for identity authentication in enormous areas such as finance, military, public security and so on. Nowadays, most face verification models are built upon Deep Convolutional Neural Networks and supervised by classification loss functions Taigman et al. (2014); Wen et al. (2016); Wang et al. (2017); Liu et al. (2017a), metric learning loss functions Schroff et al. (2015) or both Sun et al. (2014); Parkhi et al. (2015). Metric learning loss functions such as contrastive loss Sun et al. (2014) or triplet loss Schroff et al. (2015) usually require carefully designed sample mining strategies and the final performance is very sensitive to these strategies, so increasingly more researchers shift their attentions to building deep face verification models based on improved classification loss functions Wen et al. (2016); Wang et al. (2017); Liu et al. (2017a).

Current prevailing classification loss functions for deep face recognition are mostly based on the widely-used softmax loss. The softmax loss is typically good at optimizing the inter-class difference (i.e., separating different classes), but not good at reducing the intra-class variation (i.e., making features of the same class compact). To address this, lots of new loss functions are proposed to minimize the intra-class variation. Wen et al. (2016) proposed to add a regularization term to penalize the feature-to-center distances. In Wang et al. (2017); Liu et al. (2017c); Ranjan et al. (2017), researchers proposed to use a scale parameter to control the "temperature" Hinton et al. (2015) of the softmax loss, producing higher gradients to the well-separated samples to further shrink the

---

[1] `https://github.com/happynear/AMSoftmax`

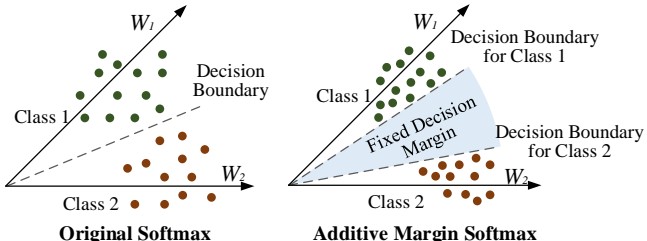

Figure 1: Comparison between the original softmax loss and the additive margin softmax loss. Note that, the angular softmax Liu et al. (2017a) can only impose unfixed angular margin, while the additive margin softmax incorporates the fixed hard angular margin.

intra-class variance. In Liu et al. (2017a; 2016), the authors introduced an conceptually appealing angular margin to push the classification boundary closer to the weight vector of each class. Liu et al. (2017a) also provided a theoretical guidance of training a deep model for metric learning tasks using the classification loss functions. Liang et al. (2017); Liu et al. (2017c); Ranjan et al. (2017) also improved the softmax loss by incorporating differnet kinds of margins.

In this work, we propose a novel and more interpretable way to import the angular margin into the softmax loss. We formulate an additive margin via $\cos\theta - m$, which is simpler than Liu et al. (2017a) and yields better performance. From Equation (3), we can see that $m$ is multiplied to the target angle $\theta_{y_i}$ in Liu et al. (2017a), so this type of margin is incorporated in a multiplicative manner. Since our margin is a scalar subtracted from $cos\theta$, we call our loss function Additive Margin Softmax (AM-Softmax).

Experiments on LFW BLUFR protocol Liao et al. (2014) and MegaFace Kemelmacher-Shlizerman et al. (2016) show that our loss function with the same network architecture achieves better results than the current state-of-the-art approaches.

## 2 PRELIMINARIES

To better understand the proposed AM-Softmax loss, we will first give a brief review of the original softmax and the A-softmax loss Liu et al. (2017a). The formulation of the original softmax loss is given by

$$
\begin{aligned}
\mathcal{L}_S &= -\frac{1}{n}\sum_{i=1}^{n} log\frac{e^{W_{y_i}^T \boldsymbol{f}_i}}{\sum_{j=1}^{c} e^{W_j^T \boldsymbol{f}_i}} \\
&= -\frac{1}{n}\sum_{i=1}^{n} log\frac{e^{\|W_{y_i}\|\|\boldsymbol{f}_i\|cos(\theta_{y_i})}}{\sum_{j=1}^{c} e^{\|W_j\|\|\boldsymbol{f}_i\|cos(\theta_j)}},
\end{aligned}
\tag{1}
$$

where $\boldsymbol{f}$ is the input of the last fully connected layer ($\boldsymbol{f}_i$ denotes the the $i$-th sample), $W_j$ is the $j$-th column of the last fully connected layer. The $W_{y_i}^T \boldsymbol{f}_i$ is also called as the target logit Pereyra et al. (2017) of the $i$-th sample.

In the A-softmax loss, the authors proposed to normalize the weight vectors (making $\|W_i\|$ to be 1) and generalize the target logit from $\|\boldsymbol{f}_i\|cos(\theta_{y_i})$ to $\|\boldsymbol{f}_i\|\psi(\theta_{y_i})$,

$$
\mathcal{L}_{AS} = -\frac{1}{n}\sum_{i=1}^{n} log\frac{e^{\|\boldsymbol{f}_i\|\psi(\theta_{y_i})}}{e^{\|\boldsymbol{f}_i\|\psi(\theta_{y_i})} + \sum_{j=1,j\neq y_i}^{c} e^{\|\boldsymbol{f}_i\|cos(\theta_j)}},
\tag{2}
$$

where the $\psi(\theta)$ is usually a piece-wise function defined as

$$
\begin{aligned}
\psi(\theta) &= \frac{(-1)^k \cos(m\theta) - 2k + \lambda cos(\theta)}{1 + \lambda}, \\
\theta &\in [\frac{k\pi}{m}, \frac{(k+1)\pi}{m}],
\end{aligned}
\tag{3}
$$

where $m$ is usually an integer larger than 1 and $\lambda$ is a hyper-parameter to control how hard the classification boundary should be pushed. During training, the $\lambda$ is annealing from $1,000$ to a small

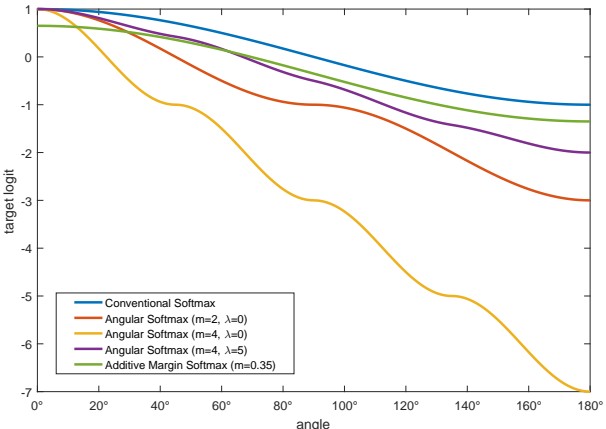

Figure 2: $\psi(\theta)$ for conventional Softmax, Angular Softmax Liu et al. (2017a) and our proposed Hard Margin Softmax. For Angular Softmax, we plot the logit curve for three parameter sets. From the curves we can infer that $m = 4, \lambda = 5$ lies between conventional Softmax and Angular Softmax with $m = 2, \lambda = 0$, which means it is approximately $m = 1.5$. Our proposed Additive Margin Softmax with optimized parameter $m = 0.35$ is also plotted and we can observe that it is similar with Angular Softmax with $m = 4, \lambda = 5$ in the range $[0°, 90°]$, in which most of the real-world $\theta$s lie.

value to make the angular space of each class become more and more compact. In their experiments, they set the minimum value of $\lambda$ to be 5 and $m = 4$, which is approximately equivalent to $m = 1.5$ (Figure 2).

## 3 ADDITIVE MARGIN SOFTMAX

In this section, we will first describe the definition of the proposed loss function. Then we will discuss about the intuition and interpretation of the loss function.

### 3.1 DEFINITION

Liu et al. (2016) defines a general function $\psi(\theta)$ to introduce the large margin property. Motivated by that, we further propose a specific $\psi(\theta)$ that introduces an additive margin to the softmax loss function. The formulation is given by

$$\psi(\theta) = cos\theta - m. \tag{4}$$

Compared to the $\psi(\theta)$ defined in L-Softmax Liu et al. (2016) and A-softmax Liu et al. (2017a) (Equation (3)), our definition is more simple and intuitive. During implementation, the input after normalizing both the feature and the weight is actually $x = cos\theta_{y_i} = \frac{W_{y_i}^T f_i}{\|W_{y_i}\|\|f_i\|}$, so in the forward propagation we only need to compute

$$\Psi(x) = x - m. \tag{5}$$

In this margin scheme, we don't need to calculate the gradient for back-propagation because $\Psi'(x) = 1$. It is much easier to implement compared with SphereFace Liu et al. (2017a).

Since we use cosine as the similarity to compare two face features, we follow Wang et al. (2017); Liu et al. (2017b;c) to apply both feature normalization and weight normalization to the inner product layer in order to build a cosine layer. Then we scale the cosine values using a hyper-parameter $s$ as

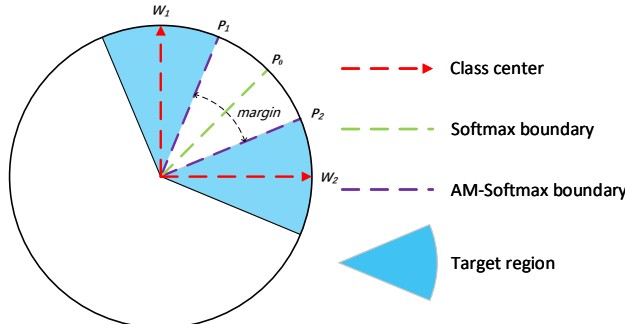

Figure 3: Conventional Softmax's decision boundary and Additive Margin Softmax's decision boundary. For conventional softmax, the decision boundary is at $P_0$, where $W_1^T P_0 = W_2^T P_0$. For AM-Softmax, the decision boundary for class 1 is at $P_1$, where $W_1^T P_1 - m = W_2^T P_1 = W_1^T P_2$. Note that the distance marked on this figure doesn't represent the real distances. The real distance is a function of the cosine of the angle, while in this figure we use the angle as the distance for better visualization effect. Here we use the word "center" to represent the weight vector of the corresponding class in the last inner-product layer, even though they may not be exactly the mean vector of the features in the class. The relationship between the weight vector ("agent") and the features' mean vector ("center") is described in Figure 6 of Wang et al. (2017).

suggested in Wang et al. (2017); Liu et al. (2017b;c). Finally, the loss function becomes

$$
\begin{aligned}
\mathcal{L}_{AMS} &= -\frac{1}{n} \sum_{i=1}^{n} log \frac{e^{s \cdot (cos\theta_{y_i} - m)}}{e^{s \cdot (cos\theta_{y_i} - m)} + \sum_{j=1, j \neq y_i}^{c} e^{s \cdot cos\theta_j}} \\
&= -\frac{1}{n} \sum_{i=1}^{n} log \frac{e^{s \cdot (W_{y_i}^T \boldsymbol{f}_i - m)}}{e^{s \cdot (W_{y_i}^T \boldsymbol{f}_i - m)} + \sum_{j=1, j \neq y_i}^{c} e^{s W_j^T \boldsymbol{f}_i}}.
\end{aligned}
\tag{6}
$$

In this paper, we assume that the norm of both $W_i$ and $\boldsymbol{f}$ are normalized to 1 if not specified. In Wang et al. (2017), the authors propose to let the scaling factor $s$ to be learned through back-propagation. However, after the margin is introduced into the loss function, we find that the $s$ will not increase and the network converges very slowly if we let $s$ to be learned. Thus, we fix $s$ to be a large enough value, e.g. 30, to accelerate and stablize the optimization.

As described in Section 2, Liu et al. (2016; 2017a) propose to use an annealing strategy to set the hyper-parameter $\lambda$ to avoid network divergence. However, to set the annealing curve of $\lambda$, lots of extra parameters are introduced, which are more or less confusing for starters. Although properly tuning those hyper-parameters for $\lambda$ could lead to impressive results, the hyper-parameters are still quite difficult to tune. With our margin scheme, we find that we no longer need the help of the annealing strategy. The network can converge flexibly even if we fix the hyper-parameter $m$ from scratch. Compared to SphereFace Liu et al. (2017a), our additive margin scheme is more friendly to those who are not familiar with the effects of the hyper-parameters. Another recently proposed additive margin is also described in Liang et al. (2017). Our AM-Softmax is different than Liang et al. (2017) in the sense that our feature and weight are normalized to a predefined constant $s$. The normalization is the key to the angular margin property. Without the normalization, the margin $m$ does not necessarily lead to large angular margin.

## 3.2 DISCUSSION

### 3.2.1 GEOMETRIC INTERPRETATION

Our additive margin scheme has a clear geometric interpretation on the hypersphere manifold. In Figure 3, we draw a schematic diagram to show the decision boundary of both conventional softmax loss and our AM-Softmax. For example, in Figure 3, the features are of 2 dimensions. After normalization, the features are on a circle and the decision boundary of the traditional softmax loss is denoted as the vector $P_0$. In this case, we have $W_1^T P_0 = W_2^T P_0$ at the decision boundary $P_0$.

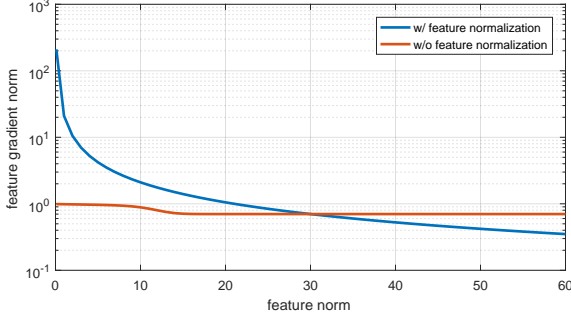

Figure 4: The feature gradient norm w.r.t. the feature norm for softmax loss with and without feature normalization. The gradients are calculated using the weights from a converged network. The feature direction is selected as the mean vector of one selected target center and one nearest class center. Note that the y-axis is in logarithmic scale for better visualization. For softmax loss with feature normalization, we set $s = 30$. That is why the intersection of these two curves is at 30.

For our AM-Softmax, the boundary becomes a marginal region instead of a single vector. At the new boundary $P_1$ for class 1, we have $W_1^T P_1 - m = W_2^T P_1$, which gives $m = (W_1 - W_2)^T P_1 = \cos(\theta_{W_1,P_1}) - \cos(\theta_{W_2,P_1})$. If we further assume that all the classes have the same intra-class variance and the boundary for class 2 is at $P_2$, we can get $\cos(\theta_{W_2,P_1}) = \cos(\theta_{W_1,P_2})$ (Fig. 3). Thus, $m = \cos(\theta_{W_1,P_1}) - \cos(\theta_{W_1,P_2})$, which is the difference of the cosine scores for class 1 between the two sides of the margin region.

### 3.2.2 Angular Margin or Cosine Margin

In SphereFace Liu et al. (2017a), the margin $m$ is multiplied to $\theta$, so the angular margin is incorporated into the loss in a multiplicative way. In our proposed loss function, the margin is enforced by subtracting $m$ from $\cos\theta$, so our margin is incorporated into the loss in an additive way, which is one of the most significant differences than Liu et al. (2017a). It is also worth mentioning that despite the difference of enforcing margin, these two types of margin formulations are also different in the base values. Specifically, one is $\theta$ and the other is $\cos\theta$. Although usually the cosine margin has an one-to-one mapping to the angular margin, there will still be some difference while optimizing them due to the non-linearity induced by the cosine function.

Whether we should use the cosine margin depends on which similarity measurement (or distance) the final loss function is optimizing. Obviously, our modified softmax loss function is optimizing the cosine similarity, not the angle. This may not be a problem if we are using the conventional softmax loss because the decision boundaries are the same in these two forms ($\cos\theta_1 = \cos\theta_2 \Rightarrow \theta_1 = \theta_2$). However, when we are trying to push the boundary, we will face a problem that these two similarities (distances) have different densities. Cosine values are more dense when the angles are near 0 or $\pi$. If we want to optimize the angle, an $\arccos$ operation may be required after the value of the inner product $W^T f$ is obtained. It will potentially be more computationally expensive.

In general, angular margin is conceptually better than the cosine margin, but considering the computational cost, cosine margin is more appealing in the sense that it could achieve the same goal with less efforts.

### 3.2.3 Feature Normalization

In the SphereFace model Liu et al. (2017a), the authors added the weight normalization based on Large Margin Softmax Liu et al. (2016), leaving the feature still not normalized. Our loss function, following Wang et al. (2017); Liu et al. (2017c); Ranjan et al. (2017), applies feature normalization and uses a global scale factor $s$ to replace the sample-dependent feature norm in SphereFace Liu et al. (2017a). One question arises: when should we add the feature normalization?

Our answer is that it depends on the image quality. In Ranjan et al. (2017)'s Figure 1, we can see that the feature norm is highly correlated with the quality of the image. Note that back propagation

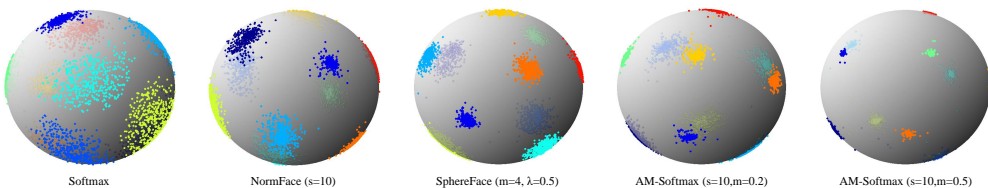

Softmax    NormFace (s=10)    SphereFace (m=4, λ=0.5)    AM-Softmax (s=10,m=0.2)    AM-Softmax (s=10,m=0.5)

Figure 5: Feature distribution visualization of several loss functions. Each point on the sphere represent one normalized feature. Different colors denote different classes. For SphereFace Liu et al. (2017a), we have already tried to use the best hyper-parameters we could find.

has a property that,

$$y = \frac{x}{\alpha} \quad \Rightarrow \quad \frac{dy}{dx} = \frac{1}{\alpha}. \tag{7}$$

Thus, after normalization, features with small norms will get much bigger gradient compared with features that have big norms (Figure 4). By back-propagation, the network will pay more attention to the low-quality face images, which usually have small norms. Its effect is very similar with hard sample mining Schroff et al. (2015); Lin et al. (2017). The advantages of feature normalization are also revealed in Liu et al. (2017b). As a conclusion, feature normalization is most suitable for tasks whose image quality is very low.

From Figure 4 we can see that the gradient norm may be extremely big when the feature norm is very small. This potentially increases the risk of gradient explosion, even though we may not come across many samples with very small feature norm. Maybe some re-weighting strategy whose feature-gradient norm curve is between the two curves in Figure 4 could potentially work better. This is an interesting topic to be studied in the future.

### 3.2.4 Feature Distribution Visualization

To better understand the effect of our loss function, we designed a toy experiment to visualize the feature distributions trained by several loss functions. We used Fashion MNIST Xiao et al. (2017) (10 classes) to train several 7-layer CNN models which output 3-dimensional features. These networks are supervised by different loss functions. After we obtain the 3-dimensional features, we normalize and plot them on a hypersphere (ball) in the 3 dimensional space (Figure 5).

From the visualization, we can empirically show that our AM-Softmax performs similarly with the best SphereFace Liu et al. (2017a) (A-Softmax) model when we set $s = 10, m = 0.2$. Moreover, our loss function can further shrink the intra-class variance by setting a larger $m$. Compared to A-Softmax Liu et al. (2017a), the AM-Softmax loss also converges easier with proper scaling factor $s$. The visualized 3D features well demonstrates that AM-Softmax could bring the large margin property to the features without tuning too many hyper-parameters.

## 4 Experiments

### 4.1 Implementation Details

Our loss function is implemented using Caffe framework (Jia et al., 2014). We follow all the experimental settings from (Liu et al., 2017a), including the image resolution, preprocessing method and the network structure. Specifically speaking, we use MTCNN (Zhang et al., 2016) to detect faces and facial landmarks in images. Then the faces are aligned according to the detected landmarks. The aligned face images are of size $112 \times 96$, and are normalized by subtracting 128 and dividing 128. Our network structure follows (Liu et al., 2017a), which is a modified ResNet (He et al., 2016) with 20 layers that is adapted to face recognition.

All the networks are trained from scratch. We set the weight decay parameter to be $5\mathrm{e}{-}4$. The batch size is 256 and the learning rate begins with 0.1 and is divided by 10 at the 16K, 24K and 28K iterations. The training is finished at 30K iterations. During training, we only use image mirror to augment the dataset. The dataset we use for training is CASIA-Webface (Yi et al., 2014), which contains 494,414 training images from 10,575 identities.

| Loss Function | $m$ | LFW BLUFR VR@FAR=0.01% | LFW BLUFR DIR@FAR=1% | MegaFace Rank1@1e6 | MegaFace VR@FAR=1e−6 |
|---|---|---|---|---|---|
| Softmax | - | 60.26% | 50.85% | 45.26% | 50.12% |
| Softmax+75% dropout | - | 77.64% | 63.72% | 57.32% | 65.58% |
| (Wen et al., 2016) | - | 83.30% | 65.46% | 63.38% | 75.68% |
| (Wang et al., 2017) | - | 88.15% | 75.22% | 65.03% | 75.88% |
| (Liu et al., 2017a) | - | 91.26% | 81.93% | 67.41% | 78.19% |
| AM-Softmax | 0.25 | 91.97% | 81.42% | 70.81% | 83.01% |
| AM-Softmax | 0.3 | 93.18% | 84.02% | 72.01% | 83.29% |
| AM-Softmax | 0.35 | 93.51% | 84.82% | **72.47%** | **84.44%** |
| AM-Softmax | 0.4 | 93.60% | 84.51% | 72.44% | 83.50% |
| AM-Softmax | 0.45 | 93.44% | 84.59% | 72.22% | 83.00% |
| AM-Softmax | 0.5 | 92.33% | 83.38% | 71.56% | 82.49% |
| AM-Softmax w/o FN | 0.35 | 93.86% | **87.58%** | 70.71% | 82.66% |
| AM-Softmax w/o FN | 0.4 | **94.48%** | 87.31% | 70.96% | 83.11% |

Table 1: Performance on modified ResNet-20 with various loss functions. Note that, for Center Loss (Wen et al., 2016) and NormFace (Wang et al., 2017), we used modified ResNet-28 (Wen et al., 2016) because we failed to train a model using Center Loss on modified ResNet-20 (Liu et al., 2017a) and the NormFace model was fine-tuned based on the Center Loss model.

To perform open-set evaluations, we carefully remove the overlapped identities between training dataset (CASIA-Webface (Yi et al., 2014)) and testing datasets (LFW (Huang et al., 2007) and MegaFace (Kemelmacher-Shlizerman et al., 2016)). In testing phase, We feed both frontal face images and mirror face images and extract the features from the output of the first inner-product layer. Then the two features are summed together as the representation of the face image. When comparing two face images, cosine similarity is utilized as the measurement.

## 4.2 DATASET OVERLAP REMOVAL

The dataset we use for training is CASIA-Webface Yi et al. (2014), which contains 494,414 training images from 10,575 identities. To perform open-set evaluations, we carefully remove the overlapped identities between training dataset (CASIA-Webface Yi et al. (2014)) and testing datasets (LFWHuang et al. (2007) and MegaFace Kemelmacher-Shlizerman et al. (2016)). Finally, we find 17 overlapped identities between CASIA-Webface and LFW, and 42 overlapped identities between CASIA-Webface and MegaFace set1. Note that there are only 80 identities in MegaFace set1, i.e. over half of the identities are already in the training dataset. The effect of overlap removal is remarkable for MegaFace (Table 4.2). To be rigorous, all the experiments in this paper are based on the cleaned dataset. We have made our overlap checking code publicly available[2] to encourage researchers to clean their training datasets before experiments.

| Loss Function | Overlap Removal? | MegaFace Rank1 | MegaFace VR |
|---|---|---|---|
| AM-Softmax | No | 75.23% | 87.06% |
| AM-Softmax | Yes | 72.47% | 84.44% |

Table 2: Effect of Overlap Removal on modified ResNet-20

In our paper, we re-train some of the previous loss functions on the cleaned dataset as the baselines for comparison. Note that, we make our experiments fair by using the same network architecture and training dataset for every compared methods.

## 4.3 EFFECT OF HYPER-PARAMETER $m$

There are two hyper-parameters in our proposed loss function, one is the scale $s$ and another is the margin $m$. The scale $s$ has already been discussed sufficiently in several previous works (Wang

---

[2]https://github.com/happynear/FaceDatasets

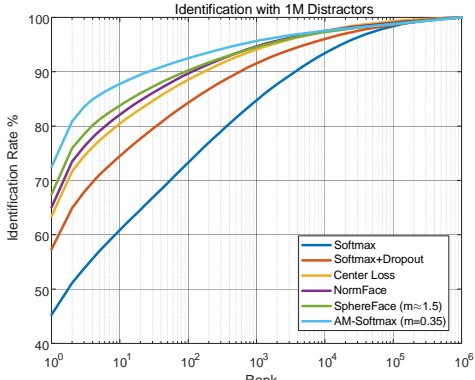 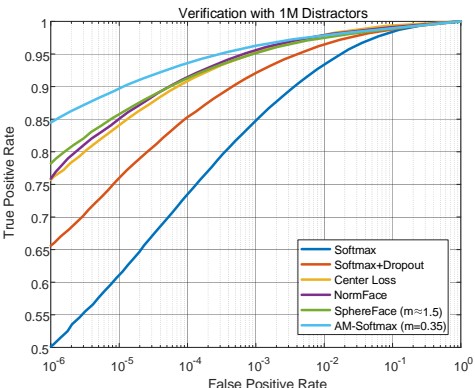

Figure 6: *Left:* CMC curves of different loss functions with 1M distractors on MegaFace (Kemelmacher-Shlizerman et al., 2016) Set 1. *Right:* ROC curves of different loss functions with 1M distractors on MegaFace (Kemelmacher-Shlizerman et al., 2016) Set 1. Note that for Center Loss and NormFace, the backend network is ResNet-28 (Wen et al., 2016), while others are based on ResNet-20(Liu et al., 2017a). Even though the curves of the Center Loss model and the NormFace model is close to the SphereFace model, please keep in mind that part of the performance comes from the bigger network structure.

et al., 2017; Liu et al., 2017c; Ranjan et al., 2017). In this paper, we directly fixed it to 30 and will not discuss its effect anymore.

The main hyper-parameter in our loss function is the margin $m$. In Table 4.1, we list the performance of our proposed AM-Softmax loss function with $m$ varies from 0.25 to 0.5. From the table we can see that from $m = 0.25$ to 0.3, the performance improves significantly, and the performance become the best when $m = 0.35$ to $m = 0.4$.

We also provide the result for the loss function without feature normalization (noted as w/o FN) and the scale $s$. As we explained before, feature normalization performs better on low quality images like MegaFace(Kemelmacher-Shlizerman et al., 2016), and using the original feature norm performs better on high quality images like LFW (Huang et al., 2007).

In Figure 6, we draw both of the CMC curves to evaluate the performance of identification and ROC curves to evaluate the performance of verification on MegaFace dataset(Kemelmacher-Shlizerman et al., 2016). From this figure, we can show that our loss function performs much better than the other loss functions when the rank or false positive rate is very low.

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
