# OpenReview forum: "Additive Margin Softmax for Face Verification"
_ICLR.cc/2018/Workshop — Accept_

### Official Review · AnonReviewer1 · 2018-03-04
**Good contribution, though it is not well-written**

**Rating:** 6
**Confidence:** 2

**Review:**

This paper proposes a simple extension method to add angular margin in a more intuitive and appealing way. While I'm not familiar with the face verification research, experimental results seem to show that their simple extensions boost the performance. Although the quality of organization and constitutions of the contents are not good, their contributions seem to be sufficient as a workshop paper.

* Pros
- Experimental settings and results are well-organized to investigate the performance of additive margin softmax.

* Cons
- The organization and written quality of the paper is not good (the detail is written below)

The organization and written quality of the paper is not good, for example,
- Explanations of formula are not sufficient. We could not interpret the definitions of many important variables, such as \theta in formula (1). The lack of these variable definitions and the reason why these formulae are constructed makes readers hard to interpret the discussion.
- The main content should not refer to the content in the appendix (such as formula (8)). If this formula is so important that the authors need to refer to this formula, this formula should be explained in the main content.

---

### Official Review · AnonReviewer2 · 2018-03-05
**This paper proposes a simple margin loss on sphere space**

**Rating:** 7
**Confidence:** 4

**Review:**

This paper proposes an additive margin loss function for the soft max loss using the cosine distance. Compared to the margin in the SphereFace, in which margin is defined by multiplied to the angular, the proposed loss is additive in original space and thus the implementation is simple.

Pros
- More easier to implement compared with ShareFace.
- The performance is better than state-of-the-art methods with the same network architecture.

Cons
-  In the Sec.4.3, the authors discussed the angular margin of SphereFace is conceptually better than the cosine margin in the proposed method. However, in the experiment, the proposed method works so well. Why the proposed angular margin works much better than SphereFace although it is not conceptually better?
-  pp.2 Equation (8) seems to be Equation (3).
- In descriptions, what means normalization is not clear. It would be better to clearly write that normalization is L2 norm normalization.
- In the descriptions of Sec.3.2 and Sec.4.4, what means low quality or high quality of image is not clear.

---

### Official Review · AnonReviewer3 · 2018-03-09
**this paper is not self-contained**

**Rating:** 5
**Confidence:** 3

**Review:**

This paper proposes a new additive margin softmax loss for face verification. Comparing with existing works, this loss is easier to understand and produces better performance.
For its quality, at first, I should say this paper is not self-contained, lots of concepts and notations are directly immigrated from previous works without explanation, or put in the appendix. These make me feel hard to judge its the contribution. For example, it’s not clear why the authors come up with this loss form before experiments.
However, its real performance looks good.  I think it’s quality will be substantially improved if the authors can provide more details.

---

### Decision · Program_Chairs · 2018-03-20
**ICLR 2018 Workshop Acceptance Decision**

**Decision:**

Accept

**Comment:**

Congratulations, your paper was accepted to the ICLR workshop.